# Evaluation of the implementation and effectiveness of a multifactorial intervention strategy for safe patient handling and movement in the healthcare sector: a study protocol of a cluster randomised controlled trial

Charlotte Wåhlin ![ORCID] [1,2] Sebastian Buck ![ORCID] [1] Jan Sandqvist ![ORCID] [3]
Paul Enthoven ![ORCID] [4] Jenni Fock,[5] Maria Andreassen ![ORCID] [3] Emma Nilsing Strid ![ORCID] [6]

For numbered affiliations see end of article.

**Correspondence to**
Dr Charlotte Wåhlin;
charlotte.wahlin@liu.se

## ABSTRACT

**Introduction** Healthcare workers with physically demanding work tasks, such as patient handling and movement (PHM), are at high risk of musculoskeletal disorders. To facilitate safe PHM and prevent musculoskeletal disorders, a combination of workplace interventions, including risk assessments, is needed. The aim of this study is to implement and evaluate a multifactorial intervention strategy for safe PHM and compare it with a single intervention strategy.

**Methods and analysis** This cluster randomised controlled trial will compare a multifactorial intervention strategy with a single intervention strategy for safe PHM in workplaces in the Swedish regional and municipal healthcare systems. At least twelve healthcare units will be recruited. Care units belonging to arm A will receive: (1) guidelines for PHM, (2) training modules, (3) risk assessment with TilThermometer, (4) risk assessment with Downtown Fall Risk Index and (5) work environment mapping. Care units belonging to Arm B will receive interventions (1) and (5). The two strategies will be evaluated with regards to (1) the primary outcome of the applied strategies' intervention effectiveness (safety climate in relation to aspects of PHM) and (2) the primary implementation outcome (acceptability, appropriateness and feasibility). This study will also explore the implementation process and intervention fidelity, examine the influence of contextual factors and investigate participants' experiences of working with strategies for safe PHM. A mix of quantitative and qualitative methods will be used. The data collection is based on questionnaires, interviews and field notes of contextual factors.

**Ethics and dissemination** The study is approved by the Swedish national ethical board (Dnr 2021–00578). Study results will be published in peer-reviewed journals, presented at conferences and distributed on social media. A lay summary and dissemination strategy will be codesigned with a reference group and participating healthcare units.

**Trial registration number** NCT05276180.

## STRENGTHS AND LIMITATIONS OF THIS STUDY

⇒ The main strengths of this multicentre project are the randomised controlled trial design with an assessment of both intervention effectiveness and implementation outcomes and the use of quantitative as well as qualitative methods.

⇒ The findings from this randomised controlled trial will contribute with knowledge about the effectiveness and experience of using two different strategies for promoting safe patient handling and movement at workplaces in healthcare settings.

⇒ Validated outcome measures are used to evaluate intervention effectiveness and implementation outcomes.

⇒ The trial findings are limited to workplaces in regional and municipal healthcare and do not include homecare.

## INTRODUCTION
### Safe healthcare environment
A safe healthcare environment is vital for both healthcare workers' (HCWs) and patients' health and safety. However, previous research indicates insufficient safety for both HCWs and patients with several risk factors that can potentially cause incidents and injuries.[1] Common occupational injuries among HCWs are those caused by sharp objects or needles; exposure to infection; injuries that occur in threatening and violent situations; falls and injuries that occur during patient handling and movement (PHM).[1–3] Musculoskeletal disorders that occur in connection with PHM are one of the most common injuries, especially in nursing staff. They are most frequent in the lumbar spine, neck and shoulders among nursing staff, especially nurses and assistant nurses.[4–7] Patient falls are

still the most frequently reported incident; however, these are reducible with multifactorial interventions and education.[8][9] Kugler *et al*[10] suggested that patient falls could be prevented with knowledge and skills in assessments of risks for patient and HCW, thereby improving safety in PHM. In addition to strategies for the active management of risks and avoiding injuries, the provision of organisational support is imperative for coping with the consequences when a patient or HCW has been injured.[11] Previous research into the relationships between HCW's working conditions and patient injuries demonstrates that a good safety climate supports safe patient care and ensures HCW safety,[12] thereby creating a safe healthcare environment. The majority of incidents and injuries among HCWs occur when caring for patients.[1] This implies a need to further focus on combining risk assessments for patients and HCWs to prevent and eliminate risks and promote a healthy, safe and sustainable healthcare environment. The importance of performing risk assessments as a basis for choosing interventions is supported by laws and regulations as well as previous research.[13][14]

### Risk assessment for safe PHM

Musculoskeletal disorders resulting from PHM are suggested as preventable by risk assessment methods that identify risk factors or evaluate work techniques.[15] One of the more comprehensive methods is TilThermometer, which focuses on the evaluation of physical exposure. The mobility level of patients and use of work equipment are correlated to the physical load on HCWs and, therefore, the level of exposure.[16][17] The TilThermometer is translated and culturally adapted from Dutch into Swedish and is available online.[18][19] In this study, we will use and evaluate the Swedish version of TilThermometer in healthcare settings. In addition, a safe PHM can also include an assessment of the patient's functioning and risk of falling.

### Interventions promoting safe PHM

Musculoskeletal disorders are not only caused by physical exposures; risk factors also include mental workload and organisational aspects, which are multifactorial and interact with each other.[6][20] This complexity must be considered when providing interventions at the individual and organisational levels. In a recent systematic review, interventions including the provision of work equipment and training HCWs were found to increase the HCWs' use of equipment, and peer coaches in safe PHM were associated with fewer occupational injuries. Other promising strategies for safe PHM were participatory ergonomics, joint management and HCW engagement.[14] However, further knowledge into how to develop a combination of workplace interventions, including risk assessments, is needed to further facilitate safe PHM and prevent musculoskeletal disorders. The intervention strategies that will be evaluated in this study are based on regulations for occupational safety and health, clinical guidelines and previous research into interventions that promote safe PHM.[13][14][21][22] These intervention strategies can be considered as complex interventions,[23] since they include several components, target both HCWs and managers, require their skills and knowledge in PHM and may demand a change in routine practice. A Swedish guideline for safe PHM was launched in December 2022 by the Swedish Agency for Work Environment Expertise for use in the healthcare sector. The goal of the guideline is to implement evidence-based practice, promote safe PHM and prevent injuries. The new Swedish guideline will be used in the present study in combination with a strategy for safe PHM at the participating care units. Further investigation into comprehensive interventions for safe PHM, the implementation process and effectiveness are warranted if we hope to improve the safety for HCWs and patients in PHM.

## AIMS AND HYPOTHESIS

The overall aim of the proposed study is to evaluate the implementation and the effectiveness of a multifactorial intervention strategy for safe PHM (A) and compare it with a single intervention strategy (B). The specific aims are as follows: (1) to evaluate and compare the intervention effectiveness of the applied strategies regarding safety climate in relation to PHM, (2) to evaluate the implementation outcome concerning the HCWs' perceptions of acceptability, appropriateness and feasibility of the intervention and (3) to explore the implementation process and fidelity to the interventions. The implementation process evaluation includes the HCWs' experiences of the multifactorial intervention strategy for safe PHM. The hypothesis in this study is that care units receiving the multifactorial intervention strategy will develop a better safety climate in relation to PHM compared with care units receiving the single intervention strategy. Furthermore, the hypothesis is that the multifactorial intervention strategy will be perceived as more acceptable, appropriate and feasible than the single intervention strategy.

## METHODS AND ANALYSIS

### Design

We propose a prospective cluster randomised controlled trial (RCT) evaluating the implementation and effectiveness of intervention strategies for safe PHM. The study is based on a hybrid 2 trial design, where clinical effectiveness and implementation outcome are given equal priority.[24] The implementation process will be evaluated using a combination of qualitative and quantitative methods, as described in the Medical Research Council guidelines for process evaluations of complex interventions.[25][26] For reporting this RCT, the study follows the Standard Protocol Items: Recommendations for interventional trials checklist.[27] The study has taken guidance from the Standards for Reporting Implementation Studies.

## Setting and participants

The study will be carried out in at least 12 healthcare units in regional and municipal healthcare centres in Sweden. At least six care units will receive the multifactorial intervention strategy (A) and at least six care units will receive the single intervention strategy (B). A care unit is defined as a unit with one manager responsible for all employees (HCWs) at the unit. A clinic or a nursing home with several smaller departments or care units will be included as one unit (cluster) to avoid contamination. We strive to recruit care units situated throughout Sweden to reach geographical spread.

## Patient and public involvement

The research project is supported by a reference group consisting of various key stakeholders, both patient organisations and unions as well as employer representatives and the Swedish work environment authority. The group was consulted during the study design and recruitment and also gave input on the guidelines for safe PHM that will be used in this study. After the trial, the trial findings will be communicated to all participating healthcare units.

## Recruitment

To recruit care units, information will be distributed during fall 2022 to workplaces in the Swedish regional and municipal healthcare systems via contact persons such as managers and human resource managers as well as other communication channels such as union newspapers, social media and newsletters. The research group will provide further information about the study and procedures to interested care units. Managers will forward information about the study and participation to the HCWs by e-mail and workplace meetings. To avoid the contamination of the different intervention strategies, no specific information about the content in the respective intervention strategies will be provided to interested care units. To explore the implementation process, a subsample of HCWs included in arm A will be invited to participate in focus group discussions or individual interviews. Information about this qualitative study will be distributed at the end of the implementation period by means of workplace meetings, emails and the project's website.

## Inclusion criteria

► Regional healthcare: inpatient care units with a minimum of 15 HCWs employed at the unit.
► Municipal healthcare: nursing homes for older adults with a minimum of 15 HCWs employed at the unit.

## Exclusion criteria

Care units providing:
► Outpatient home nursing.
► Paediatric care.
► Parts of emergency care.
► Psychiatric care.

## Procedure

As a first step, the manager will sign an informed written consent form before the care unit's participation in the study. All recruited care units willing to participate will be simultaneously randomised to either arm A or arm B by a computer-generated randomisation list. Stratification will be made in blocks by the size of the unit (large or small) and type of organisation (regional or municipal healthcare). Furthermore, care units in the same municipality and within the same regional healthcare departments will also be a factor in the stratification. The randomisation will be performed by an independent statistician prior to the baseline measurement in order to be able to adjust the planning and information to the care units belonging to arm A or arm B. The recruitment of care units and the randomisation into two intervention strategies (A and B) are presented in figure 1. The 12-month study period starts February 2023 and is divided in two phases. The first 4 month is classified as the active implementation period. In the following 8 months, the care units will continue to apply the intervention strategy.

A start-up meeting will be held by the research group in the beginning of 2023 at the included care units to introduce the strategy that the care unit has been randomised into. The HCWs at the care units will read and sign an informed written consent prior to answering any questionnaires. The cohort will be open for inclusion, allowing newly employed HCWs to enter the study.

## Intervention strategies for safe PHM

Care units will be randomised to use either a multifactorial intervention strategy (arm A) or a single intervention strategy (arm B) for safe PHM. Care units in both arm A and arm B will base their activities on the Swedish guideline for the promotion of safe PHM in their care unit during meetings, discussions and daily work. Furthermore, all care units will be encouraged to continue to work with the strategies for safe PHM, integrating them into the ordinary work at the unit until the 12-month follow-up.

Both arm A and arm B will receive a digital introduction on work environment management focusing on PHM. Arm A includes five components: (1) the Swedish guideline for PHM, a digital introduction to using it for all HCWs and in-depth strategies to discuss during two workshops for managers and implementation teams, (2) training modules (theoretical and practical), (3) risk assessment with TilThermometer (four times at each care unit),[17 19 28] (4) fall risk assessment using the Downton Fall Risk Index (DFRI)[29] and daily fall risk assessment instrument at the care unit (all patients) and (5) work environment mapping with the Structured Multidisciplinary Work Environment Survey (SMET),[30 31] including written and oral feedback to the manager. Care units randomised to arm B will receive the single intervention strategy, which contains: (1) the Swedish guideline for PHM and a digital introduction for all HCWs to using it and (2) the SMET with written feedback to manager.

## Implementation strategy

Arm A and arm B will be assigned an external facilitator who gives active support to apply the intervention strategy to the care unit during the 4-month implementation period. Furthermore, the external facilitators will be in contact with the care units up until the 12-month follow-up. The research group will give active support towards both the external facilitator and care units during the whole study period (12 months). An implementation team will be created in all care units for both arm A and arm B. The implementation team will consist of 5–6 HCWs, including the manager, a range of healthcare professions and safety representatives. The implementation team will be recommended to have regular meetings to work with their assigned intervention strategy.

In arm A, the results from the baseline SMET questionnaire will be presented as feedback, a dialogue with the manager and discussions about the systematic work environment management and conditions for safe PHM. In the same way, the results from the 4-month follow-up will be presented to the manager. Education and training in PHM will be offered to all HCWs in arm A. One session of web-based theoretical education and practical training for safe PHM will be offered to HCWs in arm A. Moreover, the implementation team and HCWs in those care units included in arm A will be able to discuss the Swedish guideline for PHM at two workshops moderated by the research group or persons appointed by the research group. Here, participants will discuss themes and the guidelines' recommendations. The implementation team will formulate several problem areas they want to work with in the workplace using a participatory approach. The TilThermometer will be used for risk assessment four times during the intervention period.

Fall risk assessment will be measured four times using DFRI (all patient's). Daily fall risk assessment is encouraged at each unit using DFRI or existing method for fall risk assessment. Arm A will be encouraged to involve their occupational healthcare services and have an active collaboration during the 4-month implementation period. The managers in arm B will receive written feedback of the results from the baseline SMET questionnaire and at the 4-month follow-up. No further implementation strategy will be given except the support from the external facilitator. Arm B will base their strategy on the Swedish guideline for safe PHM and regular meetings. When the study ends with a 12-month measurement (February 2024), the care units that have a received single intervention strategy will be given access to training concepts for safe PHM and methods for risk assessment.

## Outcome measures

A mix of quantitative and qualitative methods will be used as suggested when evaluating complex interventions.[25] The data collection will start 2023 and be based on questionnaires, interviews and field notes of contextual factors, presented in table 1.

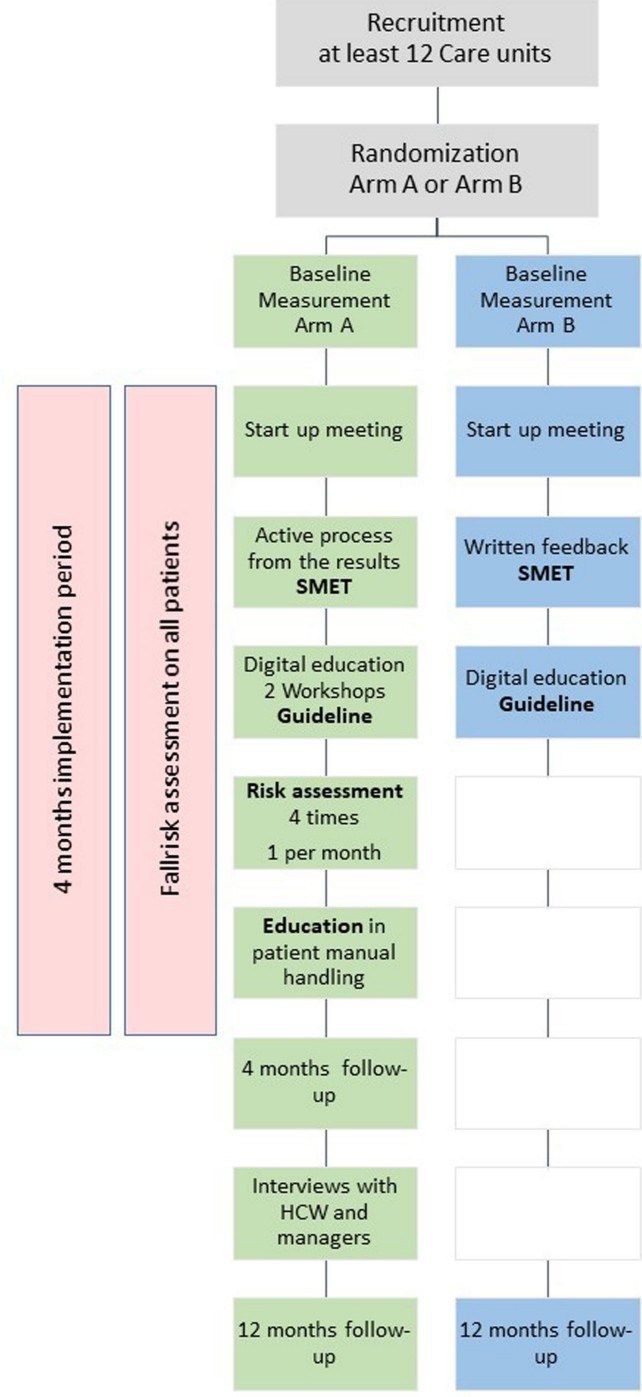

**Figure 1** Flowchart diagram showing the steps of recruitment, intervention strategies in arm A versus arm B and time point for measurement. SMET, Structured Multidisciplinary Work Environment Survey.

Questionnaires will be sent out to HCWs at participating care units and answered digitally at baseline, 4-month follow-up and 12-month follow-up. The baseline questionnaire will include background data, such as age, gender, education, working fulltime/part time, amount of overtime, profession, years of experience in the profession and current workplace.

**Table 1** Presentation of outcome measures, method of data collection, data source and time points

| Measure | Method of data collection | Data source | Time point |
|---|---|---|---|
| Primary outcome | | | |
| 1. Intervention effectiveness | | | |
| Safety climate | Questionnaire (NOSACQ-50) | HCWs and managers | T0, T1 and T2 |
| Safety for patient handling and movement | Questionnaire (research specific) | HCWs and managers | T0, T1 and T2 |
| 2. Implementation outcome | | | |
| Acceptability | Questionnaire (AIM) | HCWs and managers | T1 and T2 |
| Appropriateness | Questionnaire (IAM) | HCWs and managers | T1 and T2 |
| Feasibility | Questionnaire (FIM) | HCWs and managers | T1 and T2 |
| Secondary outcome | | | |
| 1. Intervention effectiveness | | | |
| Evaluation of the work environment | Questionnaire (SMET) | HCWs and managers | T0, T1 and T2 |
| Global Health | Questionnaire | HCWs and managers | T0, T1 and T2 |
| Work Strain | Questionnaire (Borg-CR 10) | HCWs and managers | T0, T1 and T2 |
| Work performance—work ability | Questionnaire (WAI) | HCWs and managers | T0, T1 and T2 |
| Perceived work environment Production loss Sickness presence Sickness influence on ability to work | Questionnaire | HCWs and managers | T0, T1 and T2 |
| Sickness absence | Questionnaire | HCWs and managers | T0, T1 and T2 |
| Work related musculoskeletal disorders | Questionnaire | HCWs and managers | T0, T1 and T2 |
| Fit for work—physical strength | Questionnaire (single item) | HCWs and managers | T0, T1 and T2 |
| Physical activity Daily physical activity Physical exercise the last 12 months Strenuous physical activity Time of daily physical activity | Questionnaire | HCWs and managers | T0, T1 and T2 |
| 2. Implementation outcome | | | |
| Fidelity | Observation, meeting notes, work documents, activity logs | Implementation team, external facilitator, research group | During implementation period of 4 months |
| Experiences of the intervention strategy and implementation process | Meeting notes, activity logs | Managers and implementation team members, external facilitator HCWs | Between T1 and T2 |
| Experiences of the intervention strategy and implementation process (barriers and facilitators) | Interviews: focus group and individual. CFIR will guide the analysis. | Managers and implementation team members, HCWs | Between T1 and T2 |

AIM, Acceptability of Intervention Measure; Borg CR-10, Borg Category-Ratio-10; CFIR, Consolidated Framework for Implementation Research; FIM, Feasibility of Intervention Measure; HCWs, healthcare workers ; IAM, Intervention Appropriateness Measure; NOSACQ-50, Nordic Safety Climate Questionnaire; SMET, Structured Multidisciplinary Work Environment Survey; T1, 4-month follow-up; T2, 12-month follow-up; WAI, Work Ability Index.

### Intervention effectiveness

#### Primary outcomes

The intervention effectiveness will be measured with regards to the safety climate using NOSACQ-50. This comprises 50 items across seven safety climate dimensions in the workplace (management safety priority, commitment and competence; management safety empowerment; management safety justice; workers' safety commitment; workers' safety priority and risk non-acceptance; safety communication, learning and trust in coworkers' safety competence and trust in the efficacy of safety systems).[32] The questionnaires are scored on a scale of 1–4, where 1 equals 'strongly disagree' and 4 'strongly agree', where a score greater than 3.30 indicates a good safety climate. An additional 14 specific research-based questions to evaluate the safety climate for PHM will also

be used in line with previous studies.[6] These questions involve the dimensions of using equipment, other aspects of safety, performing risk assessment and coproduction between caregivers and patients. They also use a scale of 1–4, where 1 is 'strongly disagree' and 4 is 'strongly agree', based on the design in NOSACQ-50.[32]

### Secondary outcomes

Data collected from questionnaires will include the following secondary outcomes: SMET,[30 31] which asks about physically, environmentally and psychosocially demanding work items (30 items). Another question concerns general self-reported health.[33] Musculoskeletal pain and symptoms will be measured with questions about the presence of pain and symptoms in the neck, shoulders, elbows, wrists/hands, upper back, lower back and lower extremities.[34] The Borg-CR 10 scale, specific for patient handling, movement and muscle strength,[35 36] will be used to measure perceived work strain. Individual physical strength is evaluated by the question: How do you evaluate your muscle strength in comparison to others? Data on work ability and work performance will be collected by the Work Ability Index single item as well as physical and mental demands in work, sick leave and sickness work capacity.[37 38] Furthermore, HCWs will be asked whether they have experienced any work environment problems affecting health and work performance and sickness presence in the previous 7 days.[39 40] Other questions concern the impact of sickness on the ability to work and sickness absence during the previous 12 months, for example: Is your disorder or injury affecting your ability to work? Answers range from no problems to absent due to sickness. Self-reported sickness absence has in previous studies been shown to have acceptable reliability.[41] Four questions are used to evaluate the level of physical activity.[42 43]

### Implementation outcome
#### Primary outcome

To evaluate the implementation outcome concerning the HCWs' and managers' perceptions of acceptability, appropriateness and feasibility of the intervention strategies, a set of three short questionnaires will be used.[44 45] These are validated questionnaires with the purpose of assessing the fit and match of a practice or intervention to a given context, targeting different criteria.[45] The questionnaires comprise four items each, answered on 5-point Likert-type scales. They have been translated into Swedish and cross-culturally adapted.[44] A higher score indicates a better outcome on all three parameters.

#### Secondary outcome

Fidelity is defined as the degree to which the care unit has used the intervention guideline and implemented the intervention strategy as intended by the research team.[46] To measure fidelity, a study-specific activity log will be used as recommended in previous research by Bunger *et al*.[47]

The research group will register for both arms those who attend digital introduction lectures presenting occupational safety and health for the healthcare sector and an introduction to the Swedish guideline on PHM, which is part of the intervention strategy for both arm A and B. Furthermore, the number of performed meetings with the implementation team at each care unit will be registered.

In addition, the activity log for arm A contains the five steps outlined in the intervention strategy: (1) the number of meetings and workshops carried out by the manager and implementation team, (2) the number of HCWs attending training modules, (3) the number of risk assessments with TilThermometer, (4) the percentage of fall risk assessments of all patients during the implementation period and (5) the performed SMET with active feedback through study completion. The evaluation of participating in workshops and digital and practical education will be evaluated using a questionnaire focusing on the usefulness of the content as well as how new knowledge on work environment and safe PHM can be applied in daily practice.

### Implementation process evaluation

Information about the perceived usefulness of the use of strategies for safe PHM and barriers and facilitators that may influence the implementation process will be collected by inviting managers, implementation teams and HCWs at the care units to individual interviews or focus group discussions with employers and HCWs belonging to arm A. These will be held after the 4 month implementation period and before the 12-month follow-up. An interview guide will be developed for each target group of HCWs based on the consolidated framework for implementation research (CFIR) constructs.[48]

### Contextual factors

Contextual factors will be collected from each participating care unit at baseline. These will include: the number of risk assessments the previous year; the number of managers and HCWs; the number of admissions of care recipients/places; the care burden; the number of inpatients; the amount of completed work environment training previous year; the current work environment policy; any completed training in PHM. During the whole study period, a number of explanatory and contextual factors will be collected to be able to explore the implementation process. Such factors could, for example, be organisational changes, understaffing, new managers and crises. These factors will be collected by the research team and the external facilitator in order to understand the implementation process and the outcomes of the implemented strategies in all care units. They will be collected from care units in both arm A and arm B, by the research group using field notes and conversations with managers.

## Data analysis and statistics

A power calculation was made to estimate the number of clusters and participants in each cluster. The Power calculation was based on the primary intervention effectiveness outcome NOSACQ-50, and a calculated minimum detectable change of 0.7 with an SD of 0.5 and within cluster correlation of 0.7. A power level of 0.8 (80%) and alpha of 0.05 was used. The calculation resulted in at least six care units (clusters) for each arm with a minimum of 15 participants in each care unit. Two predetermined conditions were taken into consideration in the randomisation, size of cluster and regional or municipal healthcare, as this ensured equal randomisation in arm A and arm B.

Quantitative data will be analysed using the IBM SPSS. Descriptive data are presented with mean and SD, median and range or counts and percentages. Comparisons between groups are analysed with an independent sample t test, Mann-Whitney U test or $\chi^2$ test. Multiple imputation will be used for missing data, and a sensitivity analysis will be made. Potential differences between the intervention arms regarding primary outcome at 4-month and 12-month follow-up will be estimated using linear mixed models for repeated measures. The models will be adjusted for possible confounders (baseline differences concerning age and working experience), and the results will be presented as point estimates with 95% CIs. Care units with a low rate of adherence to the assigned intervention strategy will in a subgroup analysis be compared with care units with high rate of adherence. The limit for statistical significance will be set at α=0.05. All interviews and focus group discussions will be digitally recorded and transcribed verbatim. The transcribed texts will be imported into NVivo V.12 (QSRInternational, Melbourne, Australia) to manage and code data. Qualitative content analysis will be used to analyse the data.[49] A deductive approach will be used, and the implementation framework CFIR will guide the analysis in the first step.[50] Thereafter, an inductive analysis will be used to describe the HCWs and the implementation of team members' and managers' experiences of the multifaceted intervention strategy for safe PHM with regard to the CFIR constructs in greater depth. Data from activity logs will be compiled and processed qualitatively and quantitatively.

## Ethics and dissemination

Ethics approval was received from the Swedish national ethical board (Dnr 2021–00578). HCWs working in participating care units will provide informed consent to participate in the study. It will not be possible to identify any individual study participants when the results are presented. All data in paper form, such as questionnaire responses, informed consents, code keys and transcribed interviews, will be handled by the research group with confidentiality and stored in a safe space at the research group organisation. A code list will be drawn up and stored separately from the data in a locked cabinet to which only people in the research group who are employed by Region Östergötland have access. Moreover, the results and characteristics of participating individuals will be presented at the group

level or in such a way that individuals cannot be identified. A data management plan will also be established, and all materials will be archived in accordance with current legislation and local procedures. A data monitoring committee was not needed as this is not a clinical trial with a sponsor.

**Author affiliations**
¹Occupational and Environmental Medicine Center, Department of Health, Medicine and Caring Sciences, Division of Prevention, Rehabilitation and Community Medicine, Unit of Clinical Medicine, Linköping University, Linkoping, Sweden
²Unit of Intervention and Implementation Research for worker health, Institute for Environmental Medicine, Karolinska Institutet, Stockholm, Sweden
³Department of Health, Medicine and Caring Sciences, Division of Prevention, Rehabilitation and Community Medicine, Unit of Occupational Therapy, Linköping University, Norrköping, Sweden
⁴Department of Health, Medicine and Caring Sciences, Division of Prevention, Rehabilitation and Community Medicine, Unit of Physiotherapy, Linköping University, Linköping, Sweden
⁵Unit of Stratgic Development, Linköping University Hospital, Region Östergötland, Linkoping, Sweden
⁶University Health Care Research Center, Faculty of Medicine and Health, Örebro University, Örebro, Sweden

**Acknowledgements** We would like to thank Mats Fredrikson, a statistician at Linköping University, for support with the sample size calculation and selection of appropriate statistical analysis.

**Contributors** CW is the head of the project and the cluster randomised controlled trial presented in this study protocol. CW, SB, ENS and JF will participate in the execution and collection of the project's data. CW, SB, ENS and JS are responsible for the design of the intervention strategy. All co-authors have been involved in the study design. CW, ENS and SB wrote the first draft of the manuscript, and the other co-authors (JS, JF, MA, PE) were involved in reviewing the manuscript. All authors have read and approved the final manuscript.

**Funding** This research was funded by AFA Insurance, Sweden, grant number 190144. The funders have no role or authority over the study design; collection, management, analysis or interpretation of data; writing or the decision to submit the report for publication.

**Competing interests** None declared.

**Patient and public involvement** Patients and/or the public were involved in the design, or conduct, or reporting, or dissemination plans of this research. Refer to the Methods section for further details.

**Patient consent for publication** Consent obtained directly from patient(s)

**Provenance and peer review** Not commissioned; externally peer reviewed.

**ORCID iDs**
Charlotte Wåhlin http://orcid.org/0000-0001-7847-7528
Sebastian Buck http://orcid.org/0000-0002-8925-5952
Jan Sandqvist http://orcid.org/0000-0002-9488-6142
Paul Enthoven http://orcid.org/0000-0003-3707-5869
Maria Andreassen http://orcid.org/0000-0001-8962-196X
Emma Nilsing Strid http://orcid.org/0000-0002-0483-8981

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
