## [Reviewer comments · BMJ Open]

ARTICLE DETAILS

TITLE (PROVISIONAL)	Evaluation of the Implementation and Effectiveness of a Multifactorial Intervention Strategy for Safe Patient Handling and Movement in the Healthcare Sector: A Study Protocol of a Cluster Randomised Controlled Trial
AUTHORS	Wahlin, Charlotte; Buck, Sebastian; Sandqvist, Jan; Enthoven, Paul; Fock, Jenni; Andreassen, Maria; Strid, Emma

VERSION 1 – REVIEW

REVIEWER	Vinstrup, Jonas Nationale Forskningscenter for Arbejdsmiljø
REVIEW RETURNED	09-Nov-2022

GENERAL COMMENTS	Thank you for an interesting study protocol - the upcoming intervention sounds promising. I only have a few comments: - The abstract is clearly written.- The introduction is a bit messy (and perhaps unnecessarily long), as it currently lacks a clear structure. In short, a mixture of known risk factors, Swedish laws, results from the PAWSS project, and premature mentions of methods melt together, and the aim of the study does (therefore) not follow logically. Consider shortening and re-arranging sections, and/or adding sub-headlines to make it a nicer read.-Methods: In general, a lot of information is packed into the text, making table 1 necessary in order to fully understand the aim of the study. It is clear that the authors have an in-depth understanding about how they see the future intervention proceeding, but it is difficult to achieve the same insight as an outsider reader. Consider re-arranging this section as well, to make it easier to read and understand, and to include bullet point sections, (further) explanatory tables, etc.- Please include the (intended) dates/duration of the study.- Language: Generally, the manuscript exhibits understandable written English, but could use a critical read-through by a native English speaker. There are a number of Swedish-inspired phrases and general inconsistencies that require attention; e.g. "Arm A/B" (could just be A/B, as they are mentioned a lot), numerical values, the included examples of questions should be in cursive, answer-categories more clearly defined (' '), etc. Use of punctuation needs to be consistent.
--

- According to the power calculations, >12x15 participants are needed for the intervention. As the majority of outcomes are measured using questionnaires, a significantly higher number of participants are likely to be preferred. Do you, at this time, have any idea about how many HCWs you will be able to recruit? I am asking because, as you know, recruitment of this population can be very troublesome, and the procedures proposed do not seem to fully acknowledge this issue ("information will be distributed to workplaces..." and "Several communication channels will be used to recruit units, ...").

What's in it for the wards? (especially in 'arm b'), and who is paying for time spent in training/workshops etc.?

Assuming that the shortage of HCWs is also highly prevalent in Sweden, could this mean that the wards accepting the invitation, are those who are not inherently busy? (and therefore neither representative nor free from selection-bias?).

Also, have you considered including aspects tied to the individual HCW, that might influence the perception of your outcome measures, since they are mainly based on self-reporting? (e.g. pain cognitions and fear of movement, as briefly mentioned in the introduction). I would encourage including as many as these potential risk factors / predictors as possible, as to allow subsequent analyses to identify mediators/moderators not directly related to the patient handling scenario.

- To this, as blinding is not possible, it is also likely that "arm a" will answer more positively during/following the intervention. Have you considered whether it is possible to negate this bias, and how?

Also, have you considered in/excluding different types of units/wards? Some have more patient handlings than others do, so it may be beneficial to screen for this and recruit somewhat homogenous wards.

- Since the main difference between A and B is the two risk assessments (the training aspect is negligible, and likely something most HCWs have undergone on numerous occasions), could it be argued that this is primarily what the study is evaluating? i.e. the effect of including risk assessment at the ward. To this, could you describe the possibility, if any, of more accurately pinpointing which elements of the intervention that may or may not contribute to the results of the intervention? In essence, because of the large number of influencers the different units will be exposed to during the intervention period (i.e. 'implementation teams', 'external facilitators', conversations with managers, feedback from SMET, discussions about the "systematic work environment management", two workshops, the encouragement (both A and B) to "continue to work with the strategies for safe PHM, integrating them into the ordinary work at the unit until the 12-month follow up", it would be advisable to be able to differentiate between influencers. Otherwise, the results will be of little relevance to those trying to replicate the findings.

-NOSACQ-50: Are the answer-categories correctly described? ('not correct at all' vs. 'strongly disagree')?

All in all, I think it is a decent study protocol describing what sounds to be a very promising intervention. However, the manuscript needs to be more structured and the intentions better explained and/or portrayed. Lastly, the author are encouraged to

	elaborate on how they plan to (sucessfully) recruit that many HCW (for 12 months), and how they seek to make the results of the intervention applicable outside the context of this study
--	---

REVIEWER	Jong, Gwo-Ping Chung Shan Medical University Hospital and Chung Shan Medical University, Department of Internal Medicine
-----------------	---

REVIEW RETURNED	13-Nov-2022
-------------

GENERAL COMMENTS	Charlotte et al. reported that the study protocol regarding implementing and evaluating a multifactorial intervention strategy for safe patient handling and movement (PHM) and compare it with a single intervention strategy. I think the study protocol demonstrated by the authors in this article may have clinical implications. Nevertheless, there are some issues that should be further addressed.  1. At least six care units will receive a multifactorial intervention strategy (Arm A) and at least six care units will receive a single intervention strategy (Arm B). They may be balanced in Regional healthcare and Municipal healthcare. 2. Why there is a 4-month implementation in Arm A?
---

REVIEWER	Rebbeck, Trudy Univ Sydney, Faculty of Medicine and Health
-----------------	---

REVIEW RETURNED	18-Nov-2022
-------------

GENERAL COMMENTS	Overall comments The hybrid -type II implementation effectiveness study is an appropriate design to conduct this study. The major omissions are the pre-implementation period where behaviour mapping and co-deign of the implementation strategy should occur before implementation. Using a process framework to clearly explain these steps would be helpful and is recommended in revision. Introduction The first paragraph is a over a page long, with several different constructs discussed. As such it is difficult to read. The constructs discussed include i) the risk factors that lead to work related injuries, ii) the protective factors and ii) how these could be measured. These constructs could be used as an example to divide the paragraphs up to enable easier understanding. Similar comments apply to the second long paragraph over a page long. This paragraph once again discusses several constructs whereby organisation of these constructs could be improved. These include Swedish law and its implications, interaction of factors that influence work safety amongst other things. Recommendations is to redraft the introduction into 4-5 paragraphs where each paragraph clearly focuses on a particular construct. There are many acronyms that are used – HCW, PHM, MSD’s PAWSS. Recommendation is to reduce the acronyms used, particularly when they are not common in the literature (eg HCW, PHM). Aims and Hypotheses The aims are clearly stated but the hypotheses don’t reflect the aims. For example the hypothesis for Aim 1 may be that the multifactorial intervention will be more effective for outcome A. The hypothesis for aim 2 may be that the multifactorial intervention is more acceptable and appropriate to stakeholder a,b,c.
--

	This section also then states how some outcomes will be assessed and tools that guide them, but does not explain how all outcomes are assessed. It is unusual to place this information here. Recommendation is to explain the outcomes and how they will be measured together in the methods. Methods and Analysis The cluster design and hybrid type II seems appropriate. There are a few details that are unclear that may require further clarification. These are  1) Can the efforts to avoid contamination between the clusters be explained in more detail? 2) It is unclear whether units are already recruited and have been engaged or are yet to be given this statement on the bottom of page 8 "Several communication channels will be used to recruit units, such as social media and newsletters." If not yet recruited, the authors may need to explain the feasibility of recruitment in more detail. 3) It is feasible for a manager to sign a consent form in an organisation? Would not the organisation need to be consulted and possibly ethics obtained within the organisational unit? The time frame for these governance processes may need to be detailed. 4) The intervention strategy and implementation strategy lack a co-design element bespoke for the unit. To explain, to change behaviour in any given unit, the new "behaviour" needs to be implemented in a way that is individually designed for the culture and processes of that unit. Typically the implementation strategy would be co-designed after a behaviour mapping process. This stage of implementation is lacking in this intervention. Recommendation is to add details about the behaviour mapping (or pre-implementation period) and the co-design period of the implementation strategy. 5) It is unclear who will answer the implementation outcome questionnaire. Is it all health care workers and management? 6) There are some concerns with the fidelity measure when reliant upon self report. Is a systems audit possible here? Data analysis The sample size appears low for a cluster randomised trial. How this was arrived at in relation to the primary outcome is not specified. This may require a statistician's input/ review. Please check for typographical errors – there are a few.
--	--

VERSION 1 – AUTHOR RESPONSE

Reviewer #1:

Dr. Jonas Vinstrup, Nationale Forskningscenter for Arbejdsmiljø Comments to the Author:
Thank you for an interesting study protocol - the upcoming intervention sounds promising.
I only have a few comments:

Comment 1: The abstract is clearly written. - The introduction is a bit messy (and perhaps unnecessarily long), as it currently lacks a clear structure. In short, a mixture of known risk factors, Swedish laws, results from the PAWSS project, and premature mentions of methods melt together, and the aim of the study does (therefore) not follow logically. Consider shortening and re-arranging sections, and/or adding sub-headlines to make it a nicer read.

Authors' reply: Thank you for positive feedback. We have shortened the introduction and made a new structure so that the aim becomes more logical. We have also added subheadings.

Comment 2: Methods: In general, a lot of information is packed into the text, making table 1 necessary in order to fully understand the aim of the study. It is clear that the authors have an in-depth understanding about how they see the future intervention proceeding, but it is difficult to achieve the same insight as an outsider reader. Consider re-arranging this section as well, to make it easier to read and understand, and to include bullet point sections, (further) explanatory tables, etc.

Authors' reply: We have re-arranged this section as well hoping that it will be easier to understand and follow.

Comment 3: - Please include the (intended) dates/duration of the study.

Authors' reply: Good point, this has been included.

Comment 4: - Language: Generally. The manuscript exhibits understandable written English, but could use a critical read-through by a native English speaker. There are a number of Swedish-inspired phrases and general inconsistencies that require attention; e.g. "Arm A/B" (could just be A/B, as they are mentioned a lot), numerical values, the included examples of questions should be in cursive, answer-categories more clearly defined (' '), etc. Use of punctuation needs to be consistent.

Authors' reply: We used a professional English translator. Sorry that it was not at an acceptable level. We have now used another professional translator in order to further improve the written English.

Comment 5: According to the power calculations, >12x15 participants are needed for the intervention. As the majority of outcomes are measured using questionnaires, a significantly higher number of participants are likely to be preferred. Do you, at this time, have any idea about how many HCWs you will be able to recruit?

I am asking because, as you know, recruitment of this population can be very troublesome, and the procedures proposed do not seem to fully acknowledge this issue ("information will be distributed to workplaces..." and "Several communication channels will be used to recruit units, ..."). What's in it for the wards? (especially in 'arm b'), and who is paying for time spent in training/workshops etc.?

Assuming that the shortage of HCWs is also highly prevalent in Sweden, could this mean that the wards accepting the invitation, are those who are not inherently busy? (and therefore neither representative nor free from selection-bias?).

Authors' reply: Thank you for the concern about recruitment. We are recruiting workplaces in the healthcare sector and all workplaces and there is a great interest for participating in this study. Before we have collected data at baseline it is difficult to estimate how many actually will answer the questionnaires.

We have estimated that most included workplaces have more than 30 participants, and we are expecting a minimum of 50% response rate on the questionnaire.

We have discussed the power calculation with a statistician, and we have clarified and added text concerning the power calculation: The Power calculation was based on the primary effectiveness outcome NOSACQ-50, and a calculated minimum detectable change of 0.5.

We agree that it is a challenge to recruit workplaces in this sector. However, many workplaces have shown interest, which makes us believe that the study can be carried out as planned.

Both arms are receiving a national guideline for promotion of safe patient handling and movement based on research, laws and regulations, receiving digital lectures. Both arms have assigned an implementation team (HCWs and manager) that will support the promotion of safe PHM. In Sweden, there is a great lack of updated education on PHM that has included risk assessment as well as how to apply the new knowledge with routines at the workplace. Therefore, we believe that workplaces in both intervention arms will find the new knowledge that follows with the interventions valuable.

Comment 6: Also, have you considered including aspects tied to the individual HCW, that might influence the perception of your outcome measures, since they are mainly based on self-reporting? (e.g. pain cognitions and fear of movement, as briefly mentioned in the introduction). I would encourage including as many as these potential risk factors / predictors as possible, as to allow subsequent analyses to identify mediators/moderators not directly related to the patient handling scenario. - To this, as blinding is not possible, it is also likely that "arm a" will answer more positively during/following the intervention. Have you considered whether it is possible to negate this bias, and how?

Authors' reply: In the introduction we had a reference on fear of movement, but this is not the focus of our study, we have removed this reference.

We are aware of that we are using self-reported measures for the main outcome and we agree that this can be influenced by several risk factors that can influence the outcome. Mediators and moderators are important aspects in analyzing implementation research. We will analyse mediators/moderators when doing the implementation process evaluation. In the section of Contextual factors we describe several examples of factors which will be collected during the study. Our hypothesis supports that the intervention Arm A will reach a better result compared to Arm B because of the focus on the participatory approach with workshops and a more structured intervention.

Comment 7: Also, have you considered in/excluding different types of units/wards? Some have more patient handlings than others do, so it may be beneficial to screen for this and recruit somewhat homogenous wards.

Authors' reply: This is a good point. We have excluded wards that are emergency units, psychiatric units, children's clinics, primary care settings. We have clarified that they need to have patients who are inpatients, spending at least one night at the clinic. So, we try to include rather homogeneous wards. The clinics work to promote safety according to the conditions at the clinics. All clinics will have patients of all levels of mobility, from those who can walk on their own to patients who need to be moved with a ceiling lift.

Comment 8: Since the main difference between A and B is the two risk assessments (the training aspect is negligible, and likely something most HCWs have undergone on numerous occasions), could it be argued that this is primarily what the study is evaluating? i.e. the effect of including risk assessment at the ward.

To this, could you describe the possibility, if any, of more accurately pinpointing which elements of the intervention that may or may not contribute to the results of the intervention? In essence, because of the large number of influencers the different units will be exposed to during the intervention period (i.e. 'implementation teams', 'external facilitators', conversations with managers, feedback from SMET, discussions about the "systematic work environment management", two workshops, the encouragement (both A and B) to "continue to work with the strategies for safe PHM, integrating them into the ordinary work at the unit until the 12-month follow up", it would be advisable to be able to

differentiate between influencers. Otherwise, the results will be of little relevance to those trying to replicate the findings.

Authors' reply: Thanks for your comment on the difference between A and B, we have clarified the description in the manuscript. As mentioned in the manuscript both A and B will receive the Swedish Guideline for promoting safe patient handling. Except that, units in intervention A will receive a structure on how and when they will perform risk assessment and also the analysis from it. In arm A training for all HCWs is arranged (which in Sweden is not something all HCWs have undergone, more likely not undergone any updated training). Units in intervention A will also perform workshops led by the research group or appointed persons from the research group and also discussion and guidance based on the feedback in SMET. So we argue that there is a difference between the arms. In order to increase the relevance, for those trying to replicate the study we do have a process evaluation that can be especially useful for complex and multifaceted interventions through identifying the success or failure factors of the interventions.

Comment 9: -NOSACQ-50: Are the answer-categories correctly described? ('not correct at all' vs. 'strongly disagree')?

Authors' reply: Thanks for noticing this. The correct label in English is: Strongly disagree vs. Strongly agree. This has been changed in the manuscript.

Comment 10: All in all, I think it is a decent study protocol describing what sounds to be a very promising intervention. However, the manuscript needs to be more structured and the intentions better explained and/or portrayed. Lastly, the author are encouraged to elaborate on how they plan to (successfully) recruit that many HCW (for 12 months), and how they seek to make the results of the intervention applicable outside the context of this study.

Authors' reply: Thank you for positive feedback on the planned intervention. We have tried to improve both the structure and made further explanations. We have already recruited most of the care units that are needed for this study, with an average of 40 employed healthcare workers. Our plan is to share knowledge in Sweden to healthcare units with similar care situation and also for use performing lectures and educating healthcare students on patient handling and movement.

Reviewer: 2

Dr. Gwo-Ping Jong, Chung Shan Medical University Hospital and Chung Shan Medical University, Central Taiwan University of Science and Technology

Comment 1: Charlotte et al. reported that the study protocol regarding implementing and evaluating a multifactorial intervention strategy for safe patient handling and movement (PHM) and compare it with a single intervention strategy. I think the study protocol demonstrated by the authors in this article may have clinical implications. Nevertheless, there are some issues that should be further addressed.

Authors' reply: Thank you for positive response. Yes, we do believe that the study can provide interesting clinical implications.

Comment 2: At least six care units will receive a multifactorial intervention strategy (Arm A) and at least six care units will receive a single intervention strategy (Arm B). They may be balanced in Regional healthcare and Municipal healthcare. Why there is a 4-month implementation in Arm A?

Authors' reply: We will compare the intervention strategies in arm A with arm B. We have clarified in the manuscript that both arm A and B will work with interventions during 4 months and then continue to apply their strategy up until 12 months. Thank you.

Reviewer: 3

Dr. Trudy Rebbeck, Univ Sydney

Comment 1: Overall comments. The hybrid -type II implementation effectiveness study is an appropriate design to conduct this study. The major omissions are the pre-implementation period where behaviour mapping and co-design of the implementation strategy should occur before implementation. Using a process framework to clearly explain these steps would be helpful and is recommended in revision.

Authors' reply: Thank you for supporting the design of our study and we agree with you that hybrid-type 2 is an appropriate design. We also agree that using a process framework will help us explain different outcomes in our analyses. Under Method and analysis -> design, we state the following: The implementation process will be evaluated using a combination of qualitative and quantitative methods, as described in the Medical Research Council guidelines for process evaluations of complex interventions.

We are aware of that selecting and tailoring implementation strategies according to the target groups' needs and contextual conditions are imperative for implementation success, as suggested by Damschröder et al. and other researchers. In this study, we have no pre-implementation phase, however, the target group (care unit) is involved and able to adapt the implementation strategy according to their needs.

Comment 2: Introduction- The first paragraph is a over a page long, with several different constructs discussed. As such it is difficult to read. The constructs discussed include i) the risk factors that lead to work related injuries, ii) the protective factors and ii) how these could be measured. These constructs could be used as an example to divide the paragraphs up to enable easier understanding. Similar comments apply to the second long paragraph over a page long. This paragraph once again discusses several constructs whereby organisation of these constructs could be improved. These include Swedish law and its implications, interaction of factors that influence work safety amongst other things. Recommendations is to redraft the introduction into 4-5 paragraphs where each paragraph clearly focuses on a particular construct.

Authors' reply: Thank you for useful feedback. We have re-written the introduction according to your and the first reviewers feedback.

Comment 3: There are many acronyms that are used – HCW, PHM, MSD's PAWSS. Recommendation is to reduce the acronyms used, particularly when they are not common in the literature (eg HCW, PHM).

Authors' reply: Thank you for comment on number of acronyms in our manuscript. It is easier to read without so many acronyms. We agree that it in some cases it makes it harder to grasp the content with many acronyms. However, we would like to keep Healthcare Workers (HCWs) and Patient handling and movement (PHM) since we use those several times in the manuscript. We are not using PAWSS and MSD's.

Comment 4: Aims and Hypotheses. The aims are clearly stated but the hypotheses don't reflect the aims. For example the hypothesis for Aim 1 may be that the multifactorial intervention will be more

effective for outcome A. The hypothesis for aim 2 may be that the multifactorial intervention is more acceptable and appropriate to stakeholder a,b,c.

This section also then states how some outcomes will be assessed and tools that guide them, but does not explain how all outcomes are assessed. It is unusual to place this information here. Recommendation is to explain the outcomes and how they will be measured together in the methods.

Authors' reply: Good points. We have re-structured aims and hypotheses. We have moved the information about outcomes. We have also added our hypotheses for aim 2.

Comment 5: Methods and Analysis. The cluster design and hybrid type II seems appropriate. There are a few details that are unclear that may require further clarification. These are:

1) Can the efforts to avoid contamination between the clusters be explained in more detail?

Authors' reply: Yes, we have explained this further. To avoid contamination for the different intervention strategies, no specific information about content in the intervention strategies respectively will be provided to interested units.

Comment 6: Methods and Analysis 2) It is unclear whether units are already recruited and have been engaged or are yet to be given this statement on the bottom of page 8 "Several communication channels will be used to recruit units, such as social media and newsletters." If not yet recruited, the authors may need to explain the feasibility of recruitment in more detail.

Authors' reply: Thank you for pointing this out. The units have been recruited during fall of 2022 by using several communication channels. We have clarified this in the manuscript.

Comment 7: Methods and Analysis 3) It is feasible for a manager to sign a consent form in an organisation? Would not the organisation need to be consulted and possibly ethics obtained within the organisational unit? The time frame for these governance processes may need to be detailed.

Authors' reply: According to our ethical approval the managers give consent for the units participating in the study. The care units are recruited for participation. Information about the study and what participation means will be sent out to managers of intended care units as well as information at workplace meetings or similar. The manager will sign a written consent for the care unit's participation in the study.

Comment 8: Methods and Analysis. 4) The intervention strategy and implementation strategy lack a co-design element bespoke for the unit. To explain, to change behaviour in any given unit, the new "behaviour" needs to be implemented in a way that is individually designed for the culture and processes of that unit. Typically the implementation strategy would be co-designed after a behaviour mapping process. This stage of implementation is lacking in this intervention. Recommendation is to add details about the behaviour mapping (or pre-implementation period) and the co-design period of the implementation strategy.

Authors' reply: Thank you for interesting input on a design. This is however not the design we have described in our ethical application and in the registration of clinical trials. We have developed a guide during 2022 for patient handling using methods of co-design to develop the guide that will be used and implemented. in this RCT.

As mentioned in authors reply previously we are aware of that selecting and tailoring implementation strategies according to the target groups' needs and contextual conditions are imperative for implementation success, as suggested by Damschröder et al. and other researchers. In this study, we have no pre-implementation phase, however, the target group (care unit) is involved and able to adapt the implementation strategy according to their needs.

Comment 9: Methods and Analysis. 5) It is unclear who will answer the implementation outcome questionnaire. Is it all health care workers and management?

Authors' reply: Thank you for pointing this out. We have added text to make this clear. Actually, it is both the healthcare workers and the managers. We have added text in both manuscript and table to clarify that also the managers will answer the questionnaires. However, some of the questions are not applicable for managers.

Comment 10: Methods and Analysis. 6) There are some concerns with the fidelity measure when reliant upon self-report. Is a systems audit possible here?

Authors' reply: Self-report is used as the basis for measuring fidelity during the whole study period of the 12 months. However, we do also include measures that are objective, collected by the research team, like number of HCWs attending digital and practical education, workshops, number of risk assessments (presented as implementation secondary outcomes)
System audits will be used for collecting data on a group level from each care unit concerning number and type of caretakers/patients.

Comment 11: Data analysis: The sample size appears low for a cluster randomised trial. How this was arrived at in relation to the primary outcome is not specified. This may require a statisticians input/ review.

Authors' reply: We have discussed the power calculation with a statistician, and we have clarified and added text concerning the power calculation

Comment 11: Please check for typographical errors – there are a few.

Authors' reply: Yes, we have checked for this, thank you.